

# Tunable second harmonic generation in 2D materials: Comparison of different strategies

**Simone Grillo[1]⋆, Elena Cannuccia[2], Maurizia Palummo[1],**
**Olivia Pulci[1] and Claudio Attaccalite[3,4]**

**1** University of Rome Tor Vergata, Rome, Italy
**2** Aix Marseille Univ, CNRS, PIIM, Physique des Interactions Ioniques et Moléculaires,
UMR 7345, Marseille, France
**3** CNRS/Aix-Marseille Université,
Centre Interdisciplinaire de Nanoscience de Marseille,
UMR 7325 Campus de Luminy, 13288 Marseille cedex 9, France
**4** European Theoretical Spectroscopy Facilities (ETSF)

⋆ simone.grillo1995@gmail.com

## Abstract

**Nonlinear optical frequency conversion, where optical fields interact with a nonlinear medium to generate new frequencies, is a key phenomenon in modern photonic systems. However, a major challenge with these techniques lies in the difficulty of tuning the nonlinear electrical susceptibilities that drive such effects in a given material. As a result, dynamic control of optical nonlinearities has remained largely confined to research laboratories, limiting its practical use as a spectroscopic tool. In this work, we aim to advance the development of devices with tunable nonlinear responses by exploring two potential mechanisms for electrically manipulating second-order optical nonlinearity in two-dimensional materials. Specifically, we consider two configurations: in the first, the material does not inherently exhibit second-harmonic generation (SHG), but this response is induced by an external field; in the second, an external field induces doping in a material that already exhibits SHG, altering the intensity of the nonlinear signal. In this work, we have studied these two configurations using a real-time ab-initio approach under an out-of-plane external field and including the effects of doping-induced variations in the screened electron-electron interaction. We then discuss the limitations of current computational methods and compare our results with experimental measurements.**

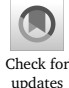

# 1 Introduction

Low dimensional materials possess a remarkable ability to respond to external environmental stimuli, such as substrates, doping, strain [1] external electric and magnetic fields [2]. These stimuli can significantly alter the opto-electronic properties of nanostructures, either bestowing new characteristics or modifying existing ones.

External electric fields, in particular, have already been used to tune various properties of low-dimensional materials, including band gaps [3], light absorption [4,5], and emission [6]. Nonlinear optical properties can also be modulated by external stimuli. The simplest nonlinear response to consider is second-harmonic generation (SHG). In SHG, a material is irradiated with a laser at a frequency $\omega$, and it emits light at a frequency $2\omega$. This phenomenon occurs only in materials lacking inversion symmetry and is highly sensitive to the angle between the laser and the crystal lattice. Due to this sensitivity, SHG has been employed, for instance, to determine the number of deposited layers [7] and their orientation [8].

Recently, several experimental groups have increasingly focused on harnessing external electric fields to either induce or enhance second-harmonic generation (SHG) in 2D materials. Two powerful strategies have emerged. The first one involves applying an electric field along the z-direction to actively induce a SHG response in materials that would typically exhibit inversion symmetry [9–11], breaking this symmetry and unlocking new optical behaviour. The second approach centers on amplifying and fine-tuning the SHG response in 2D materials that already display it, achieved by inducing doping through an external electric field and a conductive substrate, which injects electrons or holes into the system. This method not only modifies the nonlinear optical behaviour but also opens the door to precise control of the material's opto-electronic properties, further enhancing SHG efficiency [12].

Interestingly, these approaches have been combined to induce second harmonic generation (SHG) in bilayer materials with inherent inversion symmetry [13]. In this experiment, the external field not only breaks the inversion symmetry but also induces doping that charges the two layers oppositely, thereby further amplifying the symmetry breaking. This effect is known as charge-induced SHG which allows tunable frequency doubling, where the frequency and intensity of the output light can be dynamically adjusted by varying the external field or doping levels [13].

In this work, we thoroughly investigate both key mechanisms that influence second-harmonic generation in 2D materials: inversion symmetry breaking and induced doping.

These two approaches represent fundamental pathways for manipulating the nonlinear optical properties of materials. To provide a comprehensive analysis, we focus on two well-studied 2D Transition Metal Dichalcogenides (TMDs) where experimental data are already available. For the symmetry-breaking mechanism, we examine a bilayer of $MoS_2$, a material whose inherent inversion symmetry can be disrupted by an external electric field, thereby inducing a SHG response where none previously existed. For the induced doping case, we explore a monolayer of $WSe_2$, a material that naturally exhibits SHG, but whose response can be further modified and enhanced through doping with electrons or holes supplied by a conductive substrate under the influence of an external electric field.

To capture the interaction between these materials and external stimuli, we employ a real-time computational approach. This method allows us to accurately calculate the second-harmonic generation response and observe its variation as different external factors, such as electric fields and doping levels, are applied. By exploring both inversion symmetry breaking and doping-induced modifications, this study not only deepens our understanding of how SHG can be controlled in 2D materials but also highlights the potential of these mechanisms for the design of next-generation optoelectronic devices.

The manuscript is organized as follows: in Sec. 2, we present the computational methods used to study the ground state and nonlinear responses, along with the corresponding computational details and parameters required to reproduce the results. In Sec. 3, we discuss the origin of the induced or modified SHG in 2D materials under the influence of an external electric field. Finally, in Sec. 4, we present the results and compare the different strategies for tuning SHG in relation to experimental findings.

# 2 Theoretical methods

The ground-state properties of both bilayer MoS$_2$ and monolayer WSe$_2$ were studied using density functional theory (DFT) with the Quantum-Espresso code [14]. All computational details are provided in Sec. 2.2. Calculations of the optical susceptibilities were performed using the Yambo code [15]. Quasiparticle corrections to the fundamental band gap were applied as a rigid shift to all bands [16]. Finally, the nonlinear response was calculated using the real-time approach implemented in the Yambo code, as described in the following section.

## 2.1 Nonlinear response

The non-linear susceptibilities of the low dimensional materials studied in this work have been obtained from the real-time evolution of Bloch-electrons in a uniform time-dependent electric field following the approach proposed in Ref. [17].

We solved the time-dependent Schrödinger equations using an effective Hamiltonian derived from many-body perturbation theory [17] and the coupling between electrons and the external field has been treated in length gauge using the dynamical Berry Phase following the work of Souza et al: [18]

$$i\hbar \frac{d}{dt}|v_{m\mathbf{k}}\rangle = \left(H_{\mathbf{k}}^{\mathrm{MB}} + i\mathcal{E} \cdot \tilde{\partial}_{\mathbf{k}}\right)|v_{m\mathbf{k}}\rangle, \tag{1}$$

where $|v_{m\mathbf{k}}\rangle$ is the periodic part of the Bloch states, $H_{\mathbf{k}}^{\mathrm{MB}}$ is the effective Hamiltonian and $\mathcal{E} \cdot \tilde{\partial}_{\mathbf{k}}$ describes the coupling with the external field $\mathcal{E}$ in the dipole approximation [17]. In many studies, the $k$-derivative of the time-dependent valence bands is typically expressed in the crystal momentum representation (CMR). In this framework, it consists of two terms: the generalized Berry connection and the $k$-derivative of the crystal momentum wave function

(see Appendix A of Ref. [18]). The specific form of $H_{\mathbf{k}}^{\text{MB}}$ is given below. From the evolution of $|v_{m\mathbf{k}}\rangle$ in Eq. (1) we calculate the real-time polarization $\mathbf{P}_{\parallel}(t)$ along the lattice vector $\mathbf{a}$ as [18]

$$\mathbf{P}_{\parallel}(t) = -\frac{ef|\mathbf{a}|}{2\pi\Omega_c}\text{Im} \log \prod_{\mathbf{k}}^{N_{\mathbf{k}}-1} \det S(\mathbf{k}, \mathbf{k} + \mathbf{q}; t), \tag{2}$$

where $S(\mathbf{k}, \mathbf{k} + \mathbf{q}; t)$ is the overlap between the time-dependent valence states $|v_{n\mathbf{k}}(t)\rangle$ and $|v_{m\mathbf{k}+\mathbf{q}}(t)\rangle$, $\Omega_c$ is the unit cell volume, $f$ is the spin degeneracy, $N_{\mathbf{k}}$ is the number of $\mathbf{k}$ points along the polarization direction, and $\mathbf{q} = 2\pi/(N_{\mathbf{k}}\mathbf{a})$ is the discretization of the BZ in the same direction.

The polarization can be Fourier transformed and expanded in a power series of the incident field $\mathcal{E}_j$ as:

$$\mathbf{P}_i(\omega) = \chi_{ij}^{(1)}(\omega; \omega_1)\mathcal{E}_j(\omega_1) + \chi_{ijk}^{(2)}(\omega; \omega_1, \omega_2)\mathcal{E}_j(\omega_1)\mathcal{E}_k(\omega_2) + O(\mathcal{E}^3), \tag{3}$$

where $\chi^{(i)}$ are a function of the frequencies of the perturbing fields $\omega_1, \omega_2, ..$ and the outgoing polarization $\omega$. The calculation is repeated for all $\omega$ in the desired frequency range. The Fourier analysis of the real-time polarization is performed using the YamboPy code [19].

The different correlation effects can be taken into account by constructing the corresponding effective Hamiltonian somehow like in a Lego game. The full many-body Hamiltonian, $H_{\mathbf{k}}^{MB}$, available in our calculations, the so-called TD-aGW [20] reads:

$$H_{\mathbf{k}}^{MB} \equiv H_{\mathbf{k}}^{\text{KS}} + \Delta H_{\mathbf{k}} + V_h(\mathbf{r})[\Delta\rho] + \Sigma_{\text{SEX}}[\Delta\gamma]. \tag{4}$$

$H_{\mathbf{k}}^{\text{KS}}$ represents the Hamiltonian of the unperturbed (zero-field) Kohn-Sham system, while $\Delta H_{\mathbf{k}}$ is the scissors operator applied to the Kohn-Sham eigenvalues. The term $V_h(\mathbf{r})[\Delta\rho]$ is the real-time Hartree potential while $\Sigma_{\text{SEX}}$ is the screened-exchange self-energy, which accounts for the electron-hole interaction [21]. More details can be found in the Appendix. Lastly, $\Delta\rho$ and $\Delta\gamma$ represent the variations in the density and density matrix, respectively, induced by the external field.

In our simulations, we apply two external fields: a static one along the z-direction and a time-dependent in-plane one. The static one is included directly in the Kohn-Sham Hamiltonian, at the DFT level, while the time-dependent one is introduced in Eq. (1) to calculate the non-linear response. For the in-plane field, we choose a external field along the $y$ direction and the polarization is recorded in the same direction. Notice that, in the $MoS_2$ bi-layer case, $\chi^{(2)}(\omega)$ is zero if no perpendicular static field is present, while this is not the case for the $WSe_2$ monolayer.

Then for each intensity of the perpendicular static field, we performed a series of real-time simulations at different frequencies $\omega$ to extract the $\chi^{(2)}(\omega)$. Finally, in order to obtain a SHG signal independent of the dimension of the supercell, we rescaled the calculated $\chi^{(2)}(\omega)$ by an effective thickness of bilayer $MoS_2$ and monolayer $WSe_2$ — corresponding to half of $c$ lattice parameter of the bulk structures. We refer the reader to the next section for further details.

## 2.2 Computational details

In this section, we outline the parameters used in the various computational steps to obtain the nonlinear response.

Ground-state properties of both bilayer $MoS_2$ and monolayer $WSe_2$ were studied using density functional theory (DFT) with the Quantum-Espresso code [14]. We employed the Perdew-Burke-Ernzerhof (PBE) functional [22] with scalar-relativistic optimized norm-conserving pseudopotentials from the PseudoDojo repository (v0.4) [23]. Atomic structures

Table 1: This table presents all the parameters used in the linear and nonlinear response calculations for both bilayer $MoS_2$ and monolayer $WSe_2$. It includes the k-point sampling, the range of bands considered, and for $WSe_2$, it also specifies the double-grid approach used for calculating the doped dielectric constants. Additionally, the table provides details on the parameters used in constructing $\epsilon_{\mathbf{G},\mathbf{G}'}^{-1}(\omega = 0)$, the scissor operator $[\Delta E_{sc}]$ applied to the Kohn-Sham band structure, and the effective layer thickness used in the SHG calculations.

| System | **k**-points | Bands | $\epsilon_{\mathbf{G},\mathbf{G}'}$ size | $\epsilon_{\mathbf{G},\mathbf{G}'}$ bands | $\Delta E_{sc}$ | Eff. tick. |
|---|---|---|---|---|---|---|
| $2H\text{-}MoS_2$ | $42 \times 42$ | 19-34 | 5 Ha | 200 | 1.316 eV | 1.23 nm |
| $WSe_2$ | $21 \times 21\,(70 \times 70)$ | 19-26 | 4 Ha | 600 | 0.55 eV | 0.648 nm |

were relaxed using a cutoff of 120 eV, a k-point sampling of $32 \times 32 \times 1$, and the FIRE minimization algorithm [24], with a force convergence criterion of $10^{-5}$ atomic units. An electric field in the $z$-direction, perpendicular to the layers, was applied using a saw-like potential, as implemented in Quantum-Espresso. We ensured that the layers were situated in the region where the electric field is constant.

Optical susceptibilities were calculated using the Yambo code [15]. Quasiparticle corrections to the fundamental band gap were applied as a rigid shift to all bands. Parameters for the linear and nonlinear response are detailed in Table 1. For $WSe_2$, to accurately sample the doped configurations, we employed a double-grid approach for the dielectric constant calculation, as described in Ref. [25].

To include excitonic effects, we used the approach described in Appendix A and employed 1 and 5 G-vectors to model the excitonic response in bilayer $MoS_2$ and monolayer $WSe_2$, respectively. This is sufficient to reproduce the lowest excitons in $WSe_2$ and the C-exciton in the $MoS_2$ bilayer, which is responsible for the SHG response. For further details, refer to the discussion in the Appendix. We verified that increasing the number of bands in the calculations does not affect the spectra within the energy range considered here, as shown in Fig. S1 of the SI [26].

## 3 Inducing and tuning SHG

In this section, we delve into the two physical effects considered in this manuscript that affect the tunability of the SHG response. First, we examine the impact of an external out-of-plane electric field, then the effects of induced doping. We will discuss how each of these factors influences the second-order response functions. Numerical results will be presented in the following section.

### 3.1 Electronic and ionic contribution to the SHG

We start by discussing the general case of a system subjected to an external field, without any induced doping. The $i$-th component of the macroscopic second-order polarization can be written as:

$$P_i^{(2)}(2\omega) = \sum_{jk} \chi_{ijk}^{(2)}(-2\omega; \omega, \omega) E_j(\omega) E_k(\omega). \tag{5}$$

This term is zero for materials with inversion symmetry, such as the bilayer $MoS_2$ studied here. However, in the presence of a finite external static electric field $\Delta\mathcal{E}_l$, an additional term contributes to the response at $2\omega$:

$$P_i^{(2)}(2\omega) = \sum_{jk}\left\{\chi_{ijk}^{(2)}(-2\omega;\omega,\omega) + \frac{d\chi_{ijk}^{(2)}(\mathbf{R}(\mathcal{E}),\mathcal{E})}{d\mathcal{E}_l}\Delta\mathcal{E}_l\right\}E_j(\omega)E_k(\omega),\qquad(6)$$

where the last derivative can be split in two terms, an electronic and a ionic one:

$$\frac{d\chi_{ijk}^{(2)}(\mathbf{R}(\mathcal{E}),\mathcal{E})}{d\mathcal{E}_l} = \left.\frac{\partial\chi_{ijk}^{(2)}(\mathbf{R_0},\mathcal{E})}{\partial\mathcal{E}_l}\right|_{\mathcal{E}_l=0} + \sum_{n\alpha}\left.\frac{\partial\chi_{ijk}^{(2)}(\mathbf{R},\mathcal{E}=0)}{\partial\tau_{n\alpha}}\right|_{\mathbf{R}=\mathbf{R_0}}\frac{\partial\tau_{n\alpha}}{\partial\mathcal{E}_l},\qquad(7)$$

where $\tau_{n\alpha} = \mathbf{R}_{n\alpha} - \mathbf{R}_{0,n\alpha}$ represents the displacement of the atom $n$ in the direction $\alpha$. The first term in Eq. (7) is obtained by considering the atoms in their equilibrium positions and accounting only for the electronic contribution to the SHG induced by the external field $\Delta\mathcal{E}_l$. The second term corresponds to the contribution to the SHG arising from the displacement of the atoms in the presence of the external field. The first term is referred to as electric field-induced second harmonic (EFISH) generation and can be expressed as a third-order polarizability:

$$\left.\frac{\partial\chi_{ijk}^{(2)}(\mathbf{R_0},\mathcal{E})}{\partial\mathcal{E}_l}\right|_{\mathcal{E}_l=0} = \chi_{ijkl}^{(3)}(-2\omega;\omega,\omega,0).\qquad(8)$$

The two terms in Eq. (7) can be calculated directly using the approach described, for example, in Ref. [27]. However, since our goal is to study a system under the influence of a finite field, and given the special configuration of the 2D material that allows us to directly apply a finite electric field in the $z$-direction, we opted to evaluate the two terms numerically.

In order to evaluate the first term, we calculate the non-linear response while keeping the ions in their equilibrium positions. For the second term, we relax the atomic structure in presence of the external electric field and then calculate the SHG from the distorted structure without the external field, to highlight the ionic contribution. Finally, we discuss the combined effects of both contributions.

Notably, the applied electric field does not affect the linear response of the system at any level of theory, as demonstrated in Fig. S2 in the SI. [26] Although the field modifies the band structure by lifting the degeneracy of the valence and conduction bands, it does not alter the optical onset. Consequently, the intensity of the peaks remains essentially unchanged.

## 3.2  Induced doping and SHG

Doping a material can significantly alter its optical response in several ways. First, doping affects the electronic structure, such as causing the electronic band gap to narrow. Local or semi-local functionals often do not adequately capture this effect; instead, GW-type corrections are necessary. These corrections directly depend on doping due to changes in the dielectric constant, which influences the definition of $W$. Specifically, doping increases screening, which reduces $W$ and causes the GW gap to approach the DFT gap [28]. This reduction in the gap has been observed in various experiments [29]. Additionally, doping affects the atomic structure by altering bonding and lattice constants. However, these effects are not considered here because they depend also on the substrate, which is not included in the present calculations.

Beyond gap reduction, doping induces other effects in the optical response. The electron-hole interaction, which depends on the screening $\epsilon(\omega)$, is reduced, but this effect is offset by the gap shrinking. Consequently, the exciton peak position remains nearly constant for small

doping levels, while its intensity decreases [30]. At higher doping densities, this compensation fails, and the exciton peak position increases linearly with energy [30]. Doping also affects the availability of electron-hole transition channels due to partial filling (or emptying) of conduction (or valence) bands, leading to the Pauli blocking effect, which is significant for the lowest excitons. Furthermore, in doped systems, dynamical effects and the formation of trions can become important. However, our current real-time dynamics approach does not account for these effects; for a discussion, see Refs. [30, 31].

In summary, as the dynamical Berry phase approach is not applicable to metallic systems, in our real-time approach we will only consider doping effects through changes in the dielectric constant and hence the SEX self-energy, which is responsible for exciton formation in our real-time dynamics

$$\Sigma_{\text{SEX}}[\Delta\gamma](\mathbf{r}, \mathbf{r}', t) = W(\mathbf{r}, \mathbf{r}')\Delta\gamma(\mathbf{r}, \mathbf{r}', t), \tag{9}$$

$$W(\mathbf{r}, \mathbf{r}') = \epsilon^{-1}(\omega = 0)v(\mathbf{r} - \mathbf{r}'). \tag{10}$$

For each doping level, the screened interaction $W$ and consequently the self-energy $\Sigma_{\text{SEX}}$ are recalculated. The real-time dynamics, described by Eq. (1) to determine the corresponding SHG response, is performed. Note that the change in $W$ induces a small shift in the exciton position, which in principle should be offset by the shrinking of the gap. However, since we employed a rigid shift approximation, this compensation does not occur. Nonetheless, our primary focus is on studying the changes in the SHG intensity, and this shift is generally negligible in the final results. A detailed discussion of this aspect is provided in the next section.

## 4 Results

Here we present results for both the MoS$_2$ bilayer and the WSe$_2$ monolayer under the effect of an external electric field and the doping. The induced/modified second harmonic generation is discussed and compared with the pristine systems.

### 4.1 Bilayer MoS$_2$

We begin our discussion with the case of bilayer MoS$_2$. In Fig. 1, we present the ionic and electronic contributions to the SHG induced by an external electric field along the $z$-direction. Panel (b) displays the electronic contribution, corresponding to the first term in Eq. (7), obtained without relaxing the atomic structure, while panel (a) shows the ionic contribution, which corresponds to the second term in Eq. (7). The results are normalized to an effective thickness, as explained in Sec. 2.

We find that the ionic contribution is relatively small compared to the electronic counterpart. This small contribution can be rationalized as follows: to induce an atomic displacement, the external electric field must first cause a change in density, which, at first order, is proportional to $\delta\rho \propto \chi_z \mathbf{E}_z$. However, due to depolarization effects, the response $\chi_z$ of a bilayer in the perpendicular direction is quite small, explaining the minor ionic contribution. In other words, the mode polarity of the phonons along the $z$-direction is expected to be quite small. In contrast, the electronic contribution is much larger because the electric field directly influences the $\chi^{(3)}$, even without considering the induced-density effect. From these results, it is clear that, differently from many 3D materials [27], the ionic contribution in this case can be considered negligible. Therefore, in the following analysis, we will use only the equilibrium geometry and exclude the ionic effect on $\chi^{(2)}$.



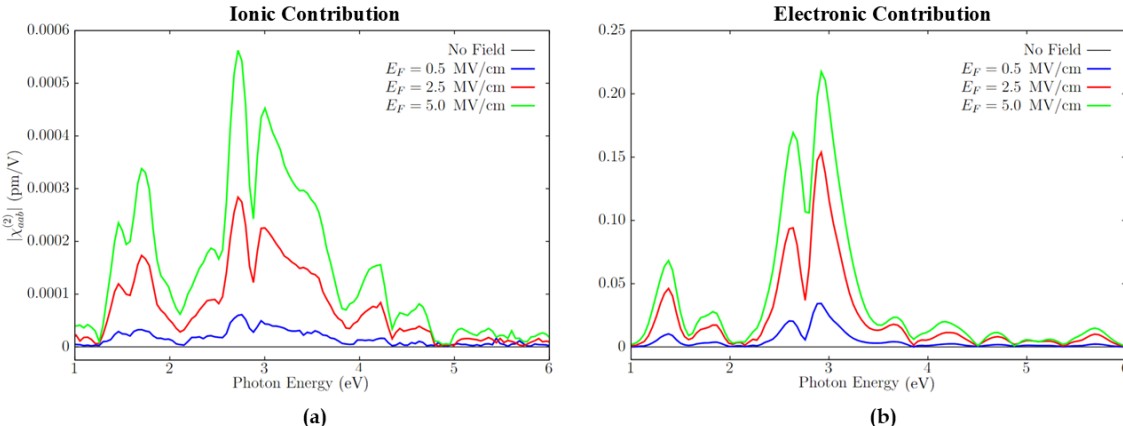

Figure 1: Ionic (**a**) and electronic (**b**) contributions to the induced SHG $\chi^{(2)}$ in bilayer MoS$_2$. In panel (**a**), we show the contribution due to the displacement of the atoms generated by the external field. Note that after atomic relaxation, the external field is turned off to consider only the ionic contribution. In panel (**b**), we present the electronic contribution to the $\chi^{(2)}$ induced by the external electric field for frozen ions.

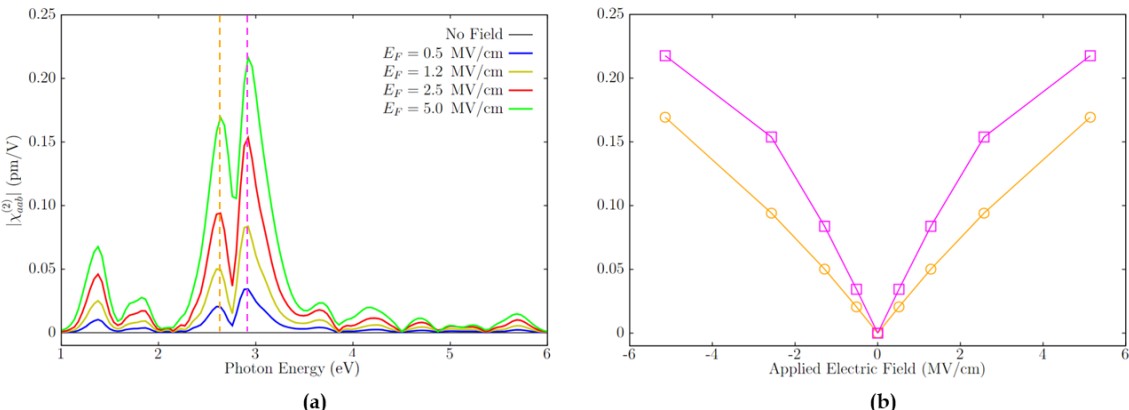

Figure 2: The two highest peaks (**b**) of the induced SHG (**a**) in bilayer MoS$_2$ as a function of the external field.

In Fig. 2, we present the two highest peaks of the induced SHG as a function of the external field. Our observations reveal a linear dependence for small fields and a quadratic dependence at higher fields, consistent with the experimental findings reported in Ref. [9]. We also attempted to incorporate excitonic effects into the SHG response using the parameters listed in Table 1. However, due to the very small signal induced by the external field, the results were quite noisy, and we decided not to include them in this manuscript. Nonetheless, even from these noisy results, we are confident that excitonic effects have the potential to more than double the SHG response in these low-dimensional systems. Recently, these effects have been investigated using a simplified model for gated bilayer graphene. Despite its relatively simple excitonic structure, this model captures many of the physical effects we observed in bilayer MoS$_2$ [32].

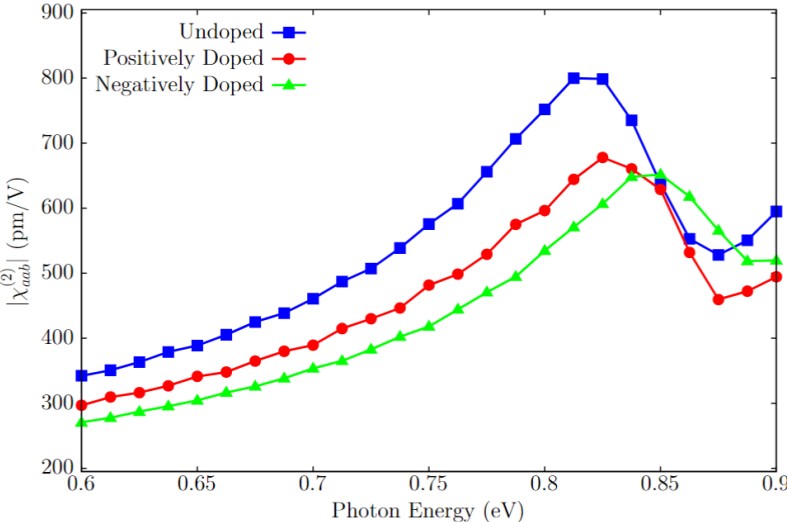

Figure 3: SHG in WSe$_2$ for the pristine undoped, positively doped with $n = 0.05e$ (holes), and negatively doped with $n = -0.05e$ (electrons) cases. Doping modifies the screened interaction $W$, Eq. (10) that enters in Eq. (4).

## 4.2 Monolayer WSe$_2$

The monolayer WSe$_2$ lacks inversion symmetry and therefore exhibits a second-harmonic signal even in the absence of external fields. However, recent studies have explored the possibility of tuning the SHG through induced doping. In its optical response, WSe$_2$ exhibits prominent bright exciton peaks that have been extensively studied both theoretically and experimentally (see Ref. [33] and references therein). The SHG response is maximized at frequencies resonant with the exciton peaks or at half of their energies. The lowest bright exciton peak in the linear response, known as the A-exciton, appears around 1.65 eV and is also observable in SHG at half this energy. Using a field-effect transistor configuration, researchers have been recetly able to modify the intensity of the A-resonance in SHG by several orders of magnitude [12].

As explained in the previous section, doping introduces two primary effects on the nonlinear response. First, the occupation of certain conduction (or valence) bands by electrons (or holes) makes these levels unavailable for valence-conduction transitions, leading to the so-called Pauli blocking effect [34]. Second, doping increases the screening of the electron-electron interaction, which reduces the excitonic resonances. This latter effect can also be achieved by depositing the material on different substrates—metallic, semiconductor, or insulator—thereby altering the dielectric constant of the system.

In this work, we focus solely on the effect of doping on the dielectric constant as we expect that this approach is equivalent to deposit the system under study on a metallic substrate.

To simulate doping, we introduced additional electron (hole) carriers along with a neutralizing uniform background charge. Specifically, we added a net charge doping of $\approx \pm 0.05e$, corresponding to a charge density of $\pm 5 \times 10^{13}$ $e/cm^2$, where $e$ is the elementary electric charge. We observed only a minor change at the DFT level, and the dielectric constant is only slightly affected by the doping. However, as we will demonstrate below, these small changes are sufficient to significantly alter the SHG response.

In Fig. 3, we present results for three cases: undoped, positively doped with $n = 0.05e$ (holes), and negatively doped with $n = -0.05e$ (electrons). We consider these extreme doping cases that could be realized in a field-effect configuration, with intermediate doping values expected to fall between these extremes. Our findings indicate that doping can diminish the SHG response at the A-resonance in WSe$_2$ by up to 20% of its intensity. This is significantly less

than the order of magnitude reduction observed experimentally [12], suggesting that other effects, such as Pauli blocking, play a crucial role in the observed reduction. This is expected since A excitons are the lowest energy states and are most affected by doping. Additionally, experiments have shown that some of the intensity may be transferred to positively and negatively charged trions, which are not included in our calculations.

Finally, we note that our calculations show a slight shift in the exciton position with doping, which is not observed in experiments. This discrepancy arises because we used a rigid shift for the QP structure. Ab initio calculations would account for this effect [30], as explained in the previous section.

## 5  Conclusions

In conclusion, this manuscript presents an ab initio study of two strategies for tuning the SHG response in 2D systems. In the first case, we investigated a system with no intrinsic SHG due to presence of inversion symmetry. An external field can induce an SHG response, which remains very weak compared to the intrinsic response of a 2D system. For this case, we found that the ionic contribution is negligible, while excitonic effects have the potential to enhance the SHG response. In the second case, we examined a system with an intrinsic SHG response and explored how doping affects its environment. We demonstrated that changing the dielectric constant can reduce the excitonic peak response up to 20%. This effect could be observed in TMD deposied on metalic substrates or in TMD/graphene heterostructures [35], which also suppresses responses from charged excitons. When we compare our result with the measurements of K. L. Seyler et al. [12], they find a reduction in the SHG response at excitonic resonance of more than 80%. However, for a correct comparison other effects should be taken into account such as Pauli blocking and the transfer of part of the spectral weight to the trions. These effects could be excluded by using heterostructure that generate a remote carrier screening without induce a charge on the material [36]. Overall, our findings illustrate how 2D materials can be employed for nonlinear optical switching. This work demonstrates the feasibility of fully ab initio simulations for predicting nonlinear optical responses in 2D materials.

## Acknowledgments

S.G, E.C. and C.A. acknowledge B. Demoulin and A. Saul for the management of the computer cluster *Rosa*.

**Funding information**   CA. and EC. acknowledge ANR project COLIBRI No. ANR-22-CE30-0027 and NOTISPERF No. ANR-20-CE47-0009-01, S.G. acknowledges the ERASMUS-PLUS program. O.P and S.G. acknowledge financial support from PRIN2020 "PHOTO". This research was partially funded by the European Union—NextGenerationEU under the Italian National Center 1 on HPC—Spoke 6: "Multiscale Modelling and Engineering Applications". O.P, M.P and S.G. thank CINECA HPC Center for computing resources through PRACE, ISCRA B and C initiatives.

## A  Approximated electron-hole interaction

In real-time simulations, time-dependent electric fields are essential for computing the time evolution of the polarization response. However, introducing a time-dependent electric field

breaks time-reversal symmetry. This presents a significant challenge, particularly for systems with a large number of electrons, such as when SOC is included, and when dense **k**-grids are used. Most computational codes for simulating periodic systems rely on symmetries to reduce computational costs and enhance performance, making the inclusion of time-dependent fields computationally demanding. Additionally, as discussed in Sec. 3 for bilayer $MoS_2$, a static electric field can induce a non-linear response, further breaking additional symmetries. This adds to the complexity of simulating such systems.

To incorporate excitonic effects into real-time dynamics, we derive a simplified screened exchange term, which we defined as long-range screened exchange (LSEX). As a starting point, we refer to the Appendix of Ref. [20]. The Kadanoff-Baym equation includes the matrix elements $\langle m, \mathbf{k} | \hat{\Sigma}^{\text{SEX}} | m', \mathbf{k} \rangle$:

$$\Sigma^{\text{SEX}}_{mm',\mathbf{k}}(t) = i \sum_{n,n'} \sum_{\mathbf{G},\mathbf{G}',\mathbf{q}} \rho_{mn}(\mathbf{k},\mathbf{q},\mathbf{G}') \rho^*_{m'n'}(\mathbf{k},\mathbf{q},\mathbf{G}) W_{\mathbf{G},\mathbf{G}'}(\mathbf{q}) \Delta\gamma_{\substack{nn'\\ \mathbf{k}-\mathbf{q}}}(t), \tag{A.1}$$

where $\Delta\gamma$ is the variation of the density matrix, and:

$$\rho_{mn}(\mathbf{k},\mathbf{q},\mathbf{G}) = \int \varphi^*_{m,\mathbf{k}}(\mathbf{r}) \varphi_{n,\mathbf{k}-\mathbf{q}}(\mathbf{r}) e^{i(\mathbf{G}+\mathbf{q})\cdot\mathbf{r}}.$$

The LSEX approximation retains only the long-range component of the screened interaction, i.e., $W(\mathbf{q}) = W_{\mathbf{G}=0,\mathbf{G}'=0}(\mathbf{q})$. Therefore, Eq. (A.1) simplifies to:

$$\Sigma^{\text{LSEX}}_{mm',\mathbf{k}}(t) = i \sum_{n,n'} \sum_{\mathbf{q}} \rho_{mn}(\mathbf{k},\mathbf{q}) \rho^*_{m'n'}(\mathbf{k},\mathbf{q}) W(\mathbf{q}) \Delta\gamma_{\substack{nn'\\ \mathbf{k}-\mathbf{q}}}(t). \tag{A.2}$$

By noting that the density matrix can be expanded as:

$$\Delta\gamma_{nm,\mathbf{k}}(t) = \sum_{l=1}^{N_v} \langle u_m | v_{l\mathbf{k}} \rangle \langle v_{l\mathbf{k}} | u_n \rangle - \delta_{nm} f(\epsilon_{n\mathbf{k}}),$$

where $l$ is an index running over valence bands, $n, n', m, m'$ are indexes running over all bands, and $f(\epsilon_{n\mathbf{k}})$ are the occupation functions, we can rewrite Eq (A.2) as:

$$\Sigma^{\text{LSEX}}_{mm',\mathbf{k}} = i \sum_{l,n,n'} \sum_{\mathbf{q}} \rho_{mn}(\mathbf{k},\mathbf{q}) \rho^*_{m'n'}(\mathbf{k},\mathbf{q}) W(\mathbf{q}) \langle u_{n'} | v_{l,\mathbf{k}-\mathbf{q}} \rangle \langle v_{l,\mathbf{k}-\mathbf{q}} | u_n \rangle - \Sigma^{\text{eq}}_{mm',\mathbf{k}}, \tag{A.3}$$

where $\Sigma^{\text{eq}}$ is the self-energy at equilibrium, defined as:

$$\Sigma^{\text{eq}}_{mm',\text{eq}} = i \sum_{n} \sum_{\mathbf{q}} \rho_{mn}(\mathbf{k},\mathbf{q}) \rho^*_{m'n'}(\mathbf{k},\mathbf{q}) W(\mathbf{q}) f(\epsilon_{n,\mathbf{k}-\mathbf{q}}).$$

We then define the oscillators $\rho$ between time-dependent valence bands and Kohn-Sham states as:

$$\tilde{\rho}_{ml}(\mathbf{k},\mathbf{q}) = \sum_{n} \rho_{mn}(\mathbf{k},\mathbf{q}) \langle v_{l,\mathbf{k}-\mathbf{q}} | u_n \rangle,$$

and Eq. (A.3) reduces to:

$$\Sigma^{\text{LSEX}}_{mm',\mathbf{k}} = i \sum_{l} \sum_{\mathbf{q}} \tilde{\rho}_{ml}(\mathbf{k},\mathbf{q}) W(\mathbf{q}) \tilde{\rho}^*_{m'l}(\mathbf{k},\mathbf{q}) - \Sigma^{\text{eq}}_{mm',\mathbf{k}}. \tag{A.4}$$

To be consistent with this approximation, we excluded the local-field effects in the dynamics. This approach is similar to that used in simple models, with the key difference that we explicitly calculate the matrix elements of the Coulomb interaction between different bands at finite **q**.

We tested this approximation on monolayer WSe$_2$ and bilayer MoS$_2$, comparing the results with available experimental measurements. We found that only a few **G**-vectors (see Eq. (A.1)) are required to reproduce the first excitons, while peaks at higher energies demand a larger number of **G**-vectors. This makes the approach less appropriate, if compared to the one in Ref. [20], for the description of the full spectrum, but very advantageous if one is interested only in the lowest excitonic peaks, see for instance Fig. S5 in SI [26].

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
