# Peer review of "Tunable second harmonic generation in 2D materials: comparison of different strategies"

_SciPost Physics Core, doi:SciPost Phys. Core 7, 081 (2024)_

## Round 1 · Referee Report · Anonymous (Referee 1) · 2024-10-30

Strengths

I have read the manuscript "Tunable second harmonic generation in 2D materials: comparison of different strategies" and, overall, I find it interesting and timely in the field of 2D materials. It presents the results of computer calculations of the non-linear (second order) susceptibility for mono- and bilayer transition metal dichalcogenides. The results provide possible routes to control the second harmonic generation in such materials.

Weaknesses

I see no significant weaknesses of the work. A minor point here is the lack of simple analytical models (that could clarify the origin of SHG, e.g., along the lines of Phys. Rev. B 95, 035311 (2017)).

Report

Overall, the manuscript is suitable for the SciPost Physics Core journal provided the authors make the requested changes, see below.

Requested changes

1.A comment regarding the validity of the "coordinate" gauge (E\cdot \partial/\partial k) is needed. In particular, the coordinate operator in the crystals contains, apart from the `i \partial/\partial k' contribution a part \Omega related to the Berry curvature of the bands. 2. The authors should clarify the origin of the resonance in \chi at 2\omega \approx 0.8..0.85 eV in monolayer WSe_2 (Fig. 3). I am unaware about any resonances in optical response of monolayers in this range. Possible, \omega \approx 0.8..0.85 eV and 2\omega \approx 1.6 .. 1.7 eV which is close to the excitonic resonances.

Recommendation

Ask for minor revision

  • validity: good
  • significance: good
  • originality: good
  • clarity: ok
  • formatting: good
  • grammar: good

Author:  Claudio Attaccalite  on 2024-11-07  [id 4943]

(in reply to Report 2 on 2024-10-30)
Category:
reply to objection
correction

Warnings issued while processing user-supplied markup:

  • Inconsistency: Markdown and reStructuredText syntaxes are mixed. Markdown will be used.
    Add "#coerce:reST" or "#coerce:plain" as the first line of your text to force reStructuredText or no markup.
    You may also contact the helpdesk if the formatting is incorrect and you are unable to edit your text.

We thank the referees for their positive feedback. Below, we provide our responses to the specific points raised by both referees and address additional comments on our manuscript:

Revisit the abstract. Title and abstract describe the setting, but not what has actually been done. At least one sentence should be added that reflects section 2 ("Theoretical Methods") of the manuscript.

In the revised version of the manuscript, we have updated the abstract to clearly describe the theoretical methods employed.

Add DOIs to all references where they are available (these are "particularly important," see https://scipost.org/SciPostPhysCore/authoring#manuprep).

We have proceeded to add DOIs to all applicable references.

Proofread titles of references, e.g., "MoS2" [7-10], "h-BN" [7], "ReS2 [11], "WSe2 [13], "GW" [16,31], "Berry" [17,18], "Bethe-Salpeter" [20,28], "Wannier" [28], "Mott" [32], "Si" [32], "Al0.7Ga0.3N" [32], and "Fröhlich" [37].

We have thoroughly proofread and corrected all reference titles to ensure accuracy and consistency.

A comment regarding the validity of the "coordinate" gauge (( E \cdot \partial/\partial k )) is needed. Specifically, the coordinate operator in crystals includes, apart from the ( i \partial/\partial k ) contribution, a term ( \Omega ) related to the Berry curvature of the bands.

We have included a sentence in the revised manuscript that explains the validity of the k-derivative of the time-dependent valence bands and its relation to the generalized Berry connection in the standard crystal momentum representation.

The authors should clarify the origin of the resonance in (\chi) at (2\omega \approx 0.8-0.85) eV in monolayer WSe(_2) (Fig. 3). I am not aware of any resonances in the optical response of monolayers in this range. Possibly, (\omega \approx 0.8-0.85) eV corresponds to (2\omega \approx 1.6-1.7) eV, which is close to excitonic resonances.

We have added a sentence explaining that the observed resonance at 0.8 eV corresponds to half the energy of the A exciton at 1.65 eV. This exciton has been confirmed in several experimental and theoretical studies. We have also added a reference where this resonance is discussed and compared with experimental data.

I see no significant weaknesses in the work. A minor point is the lack of simple analytical models (that could clarify the origin of SHG, e.g., along the lines of Phys. Rev. B 95, 035311 (2017)).

We appreciate this observation. While finalizing our manuscript, we noted the recent publication of a simplified model for nonlinear response in gated bilayer graphene in PRB. This model incorporates many of the physical effects addressed in our paper, albeit within a simpler framework than that of dichalcogenides. We have added a citation to this work in our manuscript.

---

## Round 1 · Referee Report · Anonymous (Referee 3) · 2024-11-1

Report

The authors perform first-principles calculations on the second-harmonic
generation in bilayer MoS_2 exposed to an electric field and
doped monolayer WSe_2. Notably, an approximation is used to include
electron-hole attraction effects that are typically neglected.
Interestingly, it is shown that the SHG enhancement in 2D systems due electric fields
is essentially related to electronic structure modifications, while the
ionic contributions are minor.

The results are original, interesting and well presented. They will be of
interest to the community. I suggest publication of the manuscript as is.

Recommendation

Publish (meets expectations and criteria for this Journal)

---

## Round 2 · Referee Report · Anonymous (Referee 1) · 2024-11-8

Report

I am quite satisfied with the revisions made by the authors. The only point that remains unclear is the x-axis in Fig. 3: I am still convinced that there should be "Photon energy \omega (eV)" rather than "2\omega".

Requested changes

Correct the x-axis of Fig. 3

Recommendation

Publish (meets expectations and criteria for this Journal)

  • validity: -
  • significance: -
  • originality: -
  • clarity: -
  • formatting: -
  • grammar: -

Author:  Claudio Attaccalite  on 2024-11-08  [id 4949]

(in reply to Report 1 on 2024-11-08)
Category:
reply to objection
correction

We thank the referee for this point, he/she is right, the caption of the figure is wrong . We will fix it by changing 2\omega in \omega.

---

## Round 2 · Author Response

Dear Editor

Thank you for sending us the two referee reports.
In this new version of the manuscript, we have responded to all of the referee's comments and remarks,
and to the additional comment on our publication.
All changes to the manuscript are highlighted in red.
We hope that the manuscript is now suitable for publication on SciPost Core

with best regards
Claudio Attaccalite

---

## Round 2 · List of Changes

Here is the list of changes to our manuscript:

1) We have added DOI for all references 2) We have corrected names in reference titles 3) We have added a new reference to an analytical model: Ref. 32 4) We have commented on the k derivative in Eq. 1 and its relation to the Berry connection. 5) A discussion of the A excitonic peak in WSe2 has been added with a new citation: Ref. 33 6) We have revised the abstract according to the comment on our manuscript.

---

## Editorial Decision

published